# The evolution of adhesiveness as a social adaptation

Thomas Garcia[1][*][†], Guilhem Doulcier[2][†], Silvia De Monte[2]

[1]Institut d'écologie et des sciences de l'environnement, Université Pierre et Marie Curie, Paris, France; [2]Institut de Biologie de l'École Normale Supérieure, École Normale Supérieure, PSL Research University, Paris, France

**Abstract** Cellular adhesion is a key ingredient to sustain collective functions of microbial aggregates. Here, we investigate the evolutionary origins of adhesion and the emergence of groups of genealogically unrelated cells with a game-theoretical model. The considered adhesiveness trait is costly, continuous and affects both group formation and group-derived benefits. The formalism of adaptive dynamics reveals two evolutionary stable strategies, at each extreme on the axis of adhesiveness. We show that cohesive groups can evolve by small mutational steps, provided the population is already endowed with a minimum adhesiveness level. Assortment between more adhesive types, and in particular differential propensities to leave a fraction of individuals ungrouped at the end of the aggregation process, can compensate for the cost of increased adhesiveness. We also discuss the change in the social nature of more adhesive mutations along evolutionary trajectories, and find that altruism arises before directly beneficial behavior, despite being the most challenging form of cooperation.

**\*For correspondence:**
t_garcia99@yahoo.fr

[†]These authors contributed equally to this work

**Competing interests:** The authors declare that no competing interests exist

## 1 Introduction

Some of the most peculiar behaviors observed in biological populations arise from conflicts between individual interests and collective function. The existence of different levels of organization generates situations where a behavior that is beneficial for others is associated with individual costs, in terms of reproductive success. The ubiquity of genetically encoded traits that underpin cooperative behavior has thus long been considered an evolutionary paradox, as such traits should be purged by natural selection acting at the individual level. Such paradoxical situations appear in what are known as 'major evolutionary transitions' (*Maynard-Smith and Szathmáry, 1995*), an instance of which is the transition from autonomously replicating unicellular to multicellular organisms (*Michod and Roze, 2001*; *Wolpert and Szathmáry, 2002*; *Sachs, 2008*; *Rainey and Kerr, 2010*; *Ratcliff et al., 2012*; *Hammerschmidt et al., 2014*). Evolutionary transitions all face the possibility that a part of the population free rides and exploits the collective benefit produced by the action of others. For instance, cancerous cells that divide faster disrupt the survival of their host (*Frank, 2007*).

Mathematical models, mostly based on population genetics or game theory, have pointed out several solutions to this evolutionary conundrum. In most game-theoretical and trait-group models, collective-level processes derive from individual-level behavior, and cooperation can be sustained by mechanisms that affect either population structure or individual decisions. The former mechanisms produce assortment between cooperators, arising for instance during group formation (*Avilés, 2002*; *Fletcher and Zwick, 2004*; *Santos et al., 2006*; *Guttal and Couzin, 2010*; *Powers et al., 2011*) or due to limited dispersal (*Nowak and May, 1992*; *Pfeiffer and Bonhoeffer, 2003*). The emergent topology of interaction makes cooperation profitable on average despite cooperators being outcompeted within every group, as exemplified by the so-called Simpson's paradox (*Chuang et al., 2009*). The latter mechanisms are instead effective in random population structures and alter

**eLife digest** Throughout the living world, organisms work together in groups and help each other to survive. Indeed, multicellular organisms such as plants and animals owe their existence to cooperation. Life on Earth was initially made up of single cells, some of which evolved the ability to stick to each other and work together to form tissues and organs. However, developing the ability to adhere to other cells costs energy that could otherwise be used by the cell to ensure its own survival and proliferation. How multicellularity emerged, despite such costs, remains puzzling, in particular in groups of cells that do not share a common ancestor.

Now, Garcia, Doulcier and De Monte have produced a mathematical model that shows how large cohesive groups of cells can evolve. Over long periods of time, these groups can emerge from a population of non-adhesive cells through a series of small mutations that increase the overall adhesiveness of the cells in the group. Furthermore, the evolution of cohesive groups can arise just through the cells randomly interacting. By contrast, previous models that investigated how social groups form have tended to assume that particular cell types preferentially interact with each other.

The model also suggests that the costs associated with developing adhesiveness can be partially compensated for in groups that contain cells with different abilities to adhere to each other. This means that individual cells that do not join any groups also play a crucial role in the development of cohesive groups. Finally, Garcia, Doulcier and De Monte challenge the popular belief that social behavior arises primarily because it is beneficial to the individual performing those actions. Instead, the model suggests that selfless cooperation may occur first, and only afterwards lead to the evolution of behavior that is mutually beneficial for the individuals involved.

In the future, the plausibility of the evolutionary path suggested by the model could be tested in experiments using single-celled organisms such as some amoebae and bacteria, that, along their life cycle, alternatively live alone and in cohesive groups.

individual strategy enough to offset cooperation's cost. These mechanisms typically rely on players modifying their behavior conditionally to interactants' types. For instance, reciprocity (*Trivers, 1971*; *Nowak and Sigmund, 1998*), peer recognition (*Antal et al., 2009*) and 'green beard' mechanisms (*Brown and Buckling, 2008*) or policing (*Boyd et al., 2003*) are all means by which individuals preferentially direct cooperation toward other cooperators, or retaliate against defectors. Such mechanisms require players to collect and interpret information about others, and are therefore better suited to model elaborate forms of cooperation, rather than the origin of cooperative groups themselves. Ultimately, all explanations for the evolutionary success of cooperative strategies rely on the fact that individual fitness depends on the social context, whether it is shaped by the population structure or the nature of the cooperative act.

The interest in knowing which mechanisms are effective in the evolution of pristine modes of collective organization has recently spawned numerous theoretical efforts to model microbial assemblages (*Nadell et al., 2013*; *Levin, 2014*). Although capable of thriving in isolation, most unicellular species participate—in parts of their life cycle, at least—to multicellular aggregates that offer instances of 'intermediate' integration of cells into higher levels of organismality (*Smukalla et al., 2008*; *Nadell et al., 2009*; *Queller and Strassmann, 2009*; *Rainey and Kerr, 2010*; *Ratcliff et al., 2012*; *Celiker and Gore, 2013*). Microbial societies can be highly regulated, such as in the multicellular fruiting bodies of slime moulds and myxobacteria, where cells within the collective commit to a developmental program analogous to that of permanent multicellular organisms, including the soma-like 'suicide' of a part of the population. At a lesser degree of sophistication, many other forms of collectives are known in microbial organisms, ranging from colonies to biofilms (*Grosberg, 2007*; *Nadell et al., 2009*; *Niklas, 2014*). A central feature of most microbial aggregates is their ability to persist long enough to affect the survival of their composing units. Such persistence is mediated by various forms of adhesion (*Jiang et al., 1998*; *Gresham, 2013*) ensuring local cohesion between cells of the same or different kind. For instance, cells that fail to completely separate at the moment of division are selected for in regimes favoring increased aggregate size (*Ratcliff, 2013*), a feature reinforced by genetic homogeneity within groups (*Nadell et al., 2010*). Although apparently an

achievable initial step to foster multicellularity (*Kirk, 2005*), incomplete division is however unlikely to be the only mechanism at play in the emergence of multicellular organization in the tree of life, as some biological populations form aggregates from dispersed cells and can be composed of multiple genetically distinct strains (*Nanjundiah and Sathe, 2011*). Such cases turn out to be particularly challenging for the different theoretical solutions to the evolution of collective function (*Tarnita et al., 2013*).

Here, we focus on populations where groups disperse after individual reproduction and emerge anew from a well-mixed pool at the next generation. By affecting group formation as well as the performance of a group, differential attachment is an established means to create positive assortment, which enables the concomitant evolution of adhesive traits and sizeable groups in idealized, well-mixed populations (*Simon et al., 2013*; *Garcia and De Monte, 2013*) as well as more realistic, spatially explicit settings (*Garcia et al., 2014*). These models address the binary competition between more adhesive 'cooperators' and less adhesive 'defectors' and demonstrate that segregation between types can be generated even if encounters among players occur at random. The stickier type is favored as soon as its frequency exceeds a threshold (*Garcia and De Monte, 2013*), in spite of the costs associated with increased adhesion. In actual biological populations, however, social propensity needs not result from individual features that abide by a binary logic. For instance, the FLO1 gene governing flocculation in yeast is highly variable: the level of adhesion changes as a function of the number of tandem repeats within FLO1 (*Smukalla et al., 2008*). This molecular mechanism may underpin a diversity of cell-cell interaction strengths that are better described as a trait with continuous, rather than discrete, values. In this context, the process of adaptation is mathematically formalized with the theory of adaptive dynamics (*Geritz et al., 1998*; *Waxman and Gavrilets, 2005*), describing evolution as a mutation-substitution process where mutants successively invade a monomorphic resident population. Since mutants are each time initially rare, the aforementioned frequency threshold can potentially hinder the evolution of stickiness as a gradually increasing trait.

In the following, we show that, even in the absence of selective pressures favoring large groups, natural selection can promote and sustain costly adhesion nearly from scratch, and we discuss the theoretical and applied implications of our mechanism in puzzling out the evolutionary origins of social behavior.

## 2 Adhesiveness as a quantitative trait affecting group formation and function

### 2.1 Life cycle and population structuring

We study the evolution of adhesiveness in the context, developed by *Garcia and De Monte (2013),* where the trait, associated with a fitness cost, plays a role both in the emergence of population structure—through group formation—and in the collective function of groups. Benefits derived by excreted extracellular glues, for instance, are potentially shared with neighboring individuals, while being individually costly. At the same time, glues affect the way a population clusters in groups. Here, we will consider adhesiveness as a continuous trait described by the real number $z \in [0, 1]$.

Individuals undergo successive life cycles defined by the alternation of a dispersed and a grouped phase, as illustrated in *Figure 1*. In the *Aggregation Phase* (AP), initially dispersed individuals form groups in a process that is influenced by the values of their trait. In the *Reproductive Phase* (RP), individuals leave offspring according to the benefits collected within their groups. Finally, all individuals are dispersed anew into a global pool in the *Dispersal Phase* (DP).

These assumptions, that include as a special case trait-group models with random group formation (*Wilson, 1975*), reflect actual life cycles found in species such as dictyostelids (*Li and Purugganan, 2011*; *Strassmann and Queller, 2011*) and myxobacteria (*Xavier, 2011*). Such modeled life cycle constitutes a worse-case scenario for the evolution of cooperative traits. Indeed, when aggregates are allowed to persist for more than one individual generation, positive assortment between cooperators is amplified throughout successive reproductive episodes. This can ultimately favor cooperation when population re-shuffling occurs in time scales shorter than that of defectors' takeover within groups (*Wilson, 1987*; *Fletcher and Zwick, 2004*; *Killingback et al., 2006*; *Traulsen and Nowak, 2006*; *Cremer et al., 2012*).

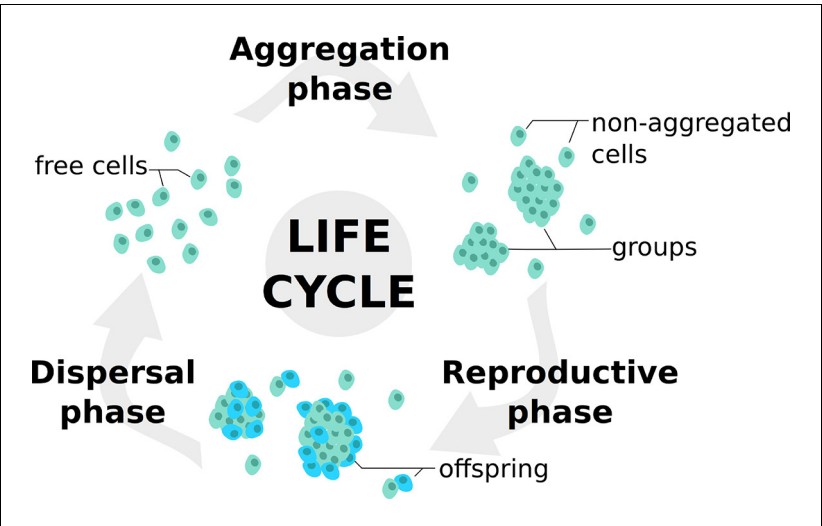

**Figure 1.** Life cycle used in the model. At each generation, individuals undergo a succession of three steps: an aggregation phase (AP) during which they form groups depending on their adhesiveness trait; a reproduction phase (RP) in which they leave offspring with a probability dependent on their strategies and their payoffs in groups; a dispersal phase (DP) when all individuals are scattered anew for the next generation. Such life cycle is consistent with those observed in facultative multicellular microorganisms such as dictyostelids and myxobacteria.

The individual trait under the effect of natural selection is adhesiveness $z$. Higher values of $z$ increase the probability that an individual joins a group (in AP), and at the same time enhance group cohesion, hence group-related benefits (in RP). Adhesiveness is costly, and the cost is assumed proportional to its value: $C(z) = c\,z$. In Section 4.4, we consider the more realistic case when the cost of glue production increases faster than linearly, for instance because of metabolic or physical constraints. The cost of adhesiveness is here assumed to be context-independent, thus it does not change conditionally to individuals belonging or not to a group. This choice reflects the standard assumption in quantitative genetics that the trait is genetically encoded. The context-dependent part is thus restricted to the benefit term. Situations where increased attachment has no cost have been modelled by *Avilés (2002)*. Assuming that adhesive ungrouped individuals do not undergo adhesion costs, or that they earn direct benefits just as grouped individuals do, would relax the social dilemma and promote even more efficiently increased adhesion.

In the RP, each individual in a group is assigned a net payoff according to a Public Goods Game (PGG) (*Kollock (1998)*; *Doebeli and Hauert (2005)*), that models in the simplest terms the reproductive success of individuals taking part in a social enterprise. The net payoff is the sum of two terms: the cost of adhesiveness, and a benefit $B$ drawn from belonging to a group, and therefore equal for all its members. This second term encapsulates the cohesion of the group and depends on the average adhesiveness $\bar{z}$ of its members: $B = B(\bar{z})$ (*Brännström et al., 2011*), with $B$ an increasing function of $\bar{z}$. In the following, we opt for a linear function $B(\bar{z}) = b\,\bar{z}$. This choice is conservative, since nonlinear (e.g. saturating) functions alleviate the constraints on the evolution of social traits (*Archetti and Scheuring, 2012*). Ultimately, a $z$-individual in a group of average adhesiveness $\bar{z}$ gets a net payoff $b\,\bar{z} - c\,z$. If an individual does not belong to any group, it does not get any group-related benefit and its payoff is merely $-c\,z$.

The average payoff of one strategy, determining its reproductive success, is obtained by averaging over all social contexts experienced, meaning that the probabilities of occurrence of each possible group composition must be known. Specifying the realized group structure in populations with multiple traits is a daunting task even under simple rules of group formation, so that the evolution of the trait $z$ can only be known by explicitly simulating the aggregation process. Since microbial populations are vast, and their interactions complex, this kind of numerical simulations can be extremely time-consuming.

If mutations on $z$ occur seldom with respect to the demographic time scale of trait substitution, however, the payoff of a mutant can be assessed in the background structure provided by the resident, monomorphic population. In this case, the realized repartition of players inside groups can be deduced from specific rules of group formation, and the framework of adaptive dynamics allows to study the gradual evolution of the trait in general settings. Section 3 discusses the adaptive dynamics of the trait in infinitely large populations where its value is associated with a given group size distribution. The general results will be applied in Section 4 to a specific aggregation model where adhesiveness underpins the probability of attachment among cells, that we introduce in the next paragraph.

## 2.2 Group formation based on attachment

The effect of adhesiveness on group formation can be exemplified by a simple model where the trait underpins physical attachment. Such model, introduced by *Garcia and De Monte (2013),* will be used later in Section 4 to illustrate the general results of Section 3.

During AP, individuals belonging to an initially dispersed, infinite population are distributed at random into patches of size $T$, akin for instance to the attraction domains observed in *Dictyostelium discoideum* aggregation (*Goldbeter, 2006*). Within every patch, one group is nucleated by a randomly drawn *recruiter*. Each focal individual among the remaining $T - 1$ is given one opportunity to attach to the recruiter, and does so with a probability that depends on both the recruiter and the focal individual's adhesivenesses. If it fails to stick to the recruiter, the individual remains alone. So as to preclude any assortment *a priori* between individuals with same adhesiveness, the probability that an individual of trait $z_1$ attaches to a recruiter of trait $z_2$ must be the geometric mean of the two adhesivenesses: $p_{\text{attach}}(z_1, z_2) = \sqrt{z_1 z_2}$ (see *Garcia and De Monte (2013)* for an explanation). Any other choice for $p_{\text{attach}}(z_1, z_2)$ entails more positive assortment of the stickier type, hence further facilitates an increase in adhesiveness. Note that this model is not spatially structured. Unlike works based on individuals playing on lattices (*Nowak and May, 1992*; *Doebeli and Hauert, 2005*; *Perc et al., 2013*) or on an explicit continuous space (e.g. *Meloni et al. (2009)*; *Chen et al. (2011)*; *Garcia et al. (2014)*), positive phenotypic assortment is not correlated with spatial proximity.

In *Garcia and De Monte (2013)*'s model, players possessed a binary attachment strategy with fixed cost, according to the standard choice of modeling 'cooperator' and 'defector' strategies. With this assumption, cooperation typically spreads once a threshold frequency of cooperators is overcome in the population, so that it is impossible for an initially rare cooperative mutant to be successful. Moreover, this threshold increases when adhesivenesses of the two types get closer, to such extent that even demographic stochasticity would be insufficient to favor mutations of small phenotypic effect. We will show that if attachment is based on a continuous trait, both these limitations are overcome.

In a monomorphic population of trait $z$, the aggregation process produces groups of various sizes (smaller than or equal to $T$) and a component of ungrouped individuals, as illustrated in *Figure 2*. Increasing the adhesiveness level $z$ leads to a rise in average group size, and a decrease in the fraction of individuals that remain ungrouped. These group size distributions define the population structure associated with a given value $z$ of adhesiveness. In the next Section, we provide a condition for increased adhesiveness to be selected, given the group size distributions that characterize a particular group formation process.

## 3 Adaptive dynamics of adhesiveness

Predicting the evolutionary outcome on adhesiveness in specific biological populations requires to determine how it influences the population structure. In this Section, we show that general conclusions can be drawn, nevertheless, from qualitative properties of the aggregation phase. Notably, the evolutionary success of adhesion critically depends on how it affects the recruitment of individuals within groups.

Let us consider a monomorphic resident population composed only of individuals of trait $\hat{z}$, challenged with the appearance of a rare mutant of trait $z = \hat{z} + dz$, where $dz$ is small. In this context, at most two trait values—resident and mutant—are present in the population at a given point in evolutionary time. Determination of average payoffs only relies on the knowledge of the distribution $g(n, z, \hat{z})$ of group sizes experienced by a $z$-mutant in a population composed of individuals with trait

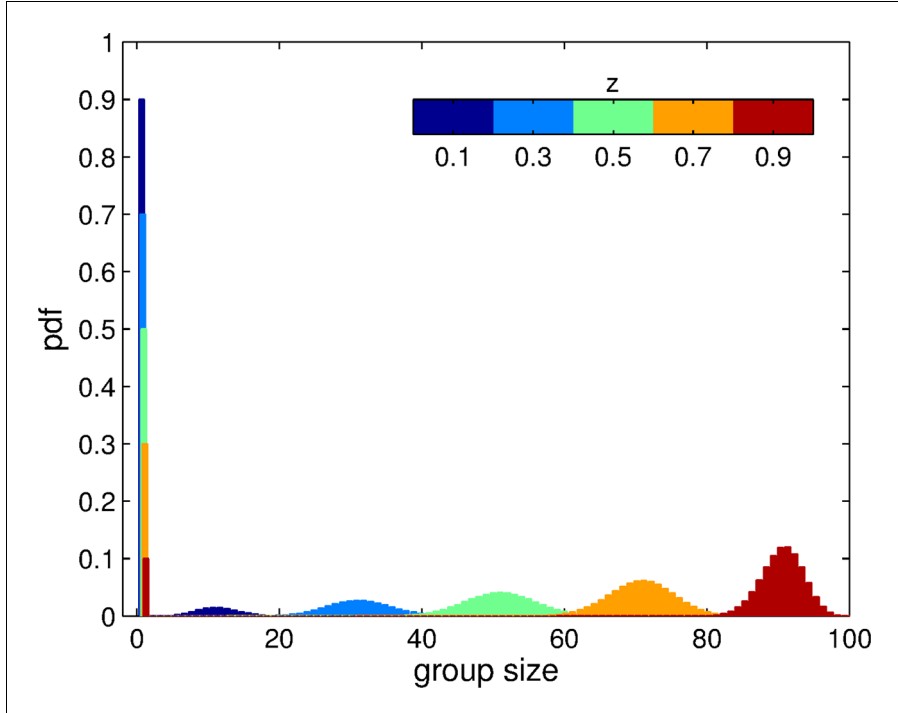

**Figure 2.** Group size distribution experienced by individuals in a momomorphic population with trait value $z$, for the aggregation process based on adhesion. The size of each patch is $T = 100$. The distribution is composed of a fraction $1 - z$ of ungrouped individuals ($n = 1$) and a binomial distribution of grouped individuals centered on $n = z\,T$. Here, we display this distribution for 5 distinct values of $z$.

$\hat{z}$, i.e. the probability that a mutant belongs to a group of size $n$. In particular, we call $u(z, \hat{z}) = g(1, z, \hat{z})$ the probability that a $z$-mutant remains alone. These distributions can be derived—analytically, numerically, or by direct observation—for any given group formation process, and summarize the effect of adhesiveness on group formation.

In line with the adaptive dynamics framework, we assume that the evolutionary dynamics of the trait $z$ is fully determined by the selection gradient (*Geritz et al., 1998*). In Section S1 of the Supplementary Information (SI), we calculate the invasion fitness of mutants $S(z, \hat{z})$ for a group formation process characterized by its group size distribution $g(n, \hat{z}, \hat{z})$ and by the proportion $u(z, \hat{z})$ of ungrouped individuals. The selection gradient is the variation of the invasion fitness with a change in adhesiveness:

$$\nabla S(\hat{z}) = \left.\frac{\partial S(z, \hat{z})}{\partial z}\right|_{z=\hat{z}} = b\,\hat{z}h(\hat{z}) + b\sum_{n\geq 2}\frac{1}{n}g(n, \hat{z}, \hat{z}) - c, \tag{1}$$

where:

$$h(\hat{z}) = -\left.\frac{\partial u(z, \hat{z})}{\partial z}\right|_{z=\hat{z}} \tag{2}$$

Since higher adhesiveness makes individuals more likely to join groups, $u(\cdot, \hat{z})$ is a decreasing function of $z$ and $h(\hat{z}) > 0$.

The change $dS$ in invasion fitness associated with an infinitesimal increase of the adhesiveness trait has three components. The third, negative term is the additional cost $-c\,dz$. The second, positive term, corresponds to direct benefits associated with the change in adhesiveness. It becomes negligible when groups are large, consistent with the observation that sociality gets established more easily in small groups (*Olson, 1971*; *Powers et al., 2011*). Even when this term cannot offset the cost of increased adhesiveness, the first component can compensate. This component

corresponds to the increased chance $h(\hat{z})\,dz$ for $z$-individuals to join a group, multiplied by the pay-off $b\,\hat{z}$ drawn from it, and thus reflects the gain associated with the new interaction neighborhood created by a change in adhesiveness.

The adaptive dynamics framework states that a more adhesive mutant $z > \hat{z}$ will replace the resident if the selection gradient in *Equation 1* is positive. A sufficient condition for an increase in adhesiveness, that holds locally for any $\hat{z}$, is then:

$$b\,\hat{z}\,h(\hat{z}) > c \qquad (3)$$

The function $h(z)$ measures the impact of adhesiveness on the fraction of cells that remain ungrouped at the end of aggregation. Depending on its variation, *Equation 3* can provide a global solution to the long-term evolutionary outcome. For instance, when $h$ increases (i.e. if $u$ is concave), or decreases slowly with $z$, then if the inequality is satisfied for one particular value of the trait, it also holds for larger values. Adhesiveness then steadily increases until it reaches its maximal value $z = 1$.

On the contrary, evolutionarily stable coexistence between grouped and ungrouped cells is only possible if $\nabla S(\hat{z})$ switches from positive to negative for some value $\hat{z} \in ]0,1[$. A necessary condition for this to happen is that the first term in *Equation 1* goes below $c$. A first possibility is that $\hat{z}$ is small, but then, arguably, groups would be small too and direct benefits large, so that the second term might keep $\nabla S(\hat{z})$ positive. A second possibility is that $h(\hat{z})$ is small, meaning that the effect of adhesiveness on the proportion of ungrouped cells is limited. This emphasizes qualitatively the importance of ungrouped individuals on the evolutionary trajectory, and how their proportion might be considered a first-order proxy of population structure.

If all individuals are randomly distributed among groups of fixed size, or even group size distribution is assigned a priori (as considered e.g. by *Peña (2012)*), the fraction of ungrouped cells is independent of $\hat{z}$. According to *Equation 3*, then, the evolution of increased adhesiveness is still possible, but it will be the consequence of direct, group-derived benefits (second term of *Equation 1*).

Life cycles consisting of alternate phases of dispersion and aggregation are more likely to leave ungrouped cells than life cycles based on single-cell bottlenecks followed by clonal growth of groups. *Equation 3* implies that a subpopulation of nonaggregated cells might boost the evolution of collective function even when cells are genetically unrelated and benefits are weak, and suggests a possible route to the evolution of multicellularity in genetically heterogeneous populations.

# 4 The evolution of adhesiveness by attachment

The results of Section 3 are valid for any group formation process taking place in the AP. In this Section, we study in detail the evolutionary dynamics of adhesiveness when groups are formed according to the rules introduced in Section 2.2.

## 4.1 Evolutionary equilibria

For any $b > 2c$, the evolutionary dynamics is bistable. The actual evolutionary outcome depends on whether the initial adhesiveness value is smaller or larger than a threshold value $z^*$, where the selection gradient vanishes. Its analytical expression is computed in Section S2 of the SI.

If initial adhesiveness is lower than $z^*$, additional group-related benefits are too small to offset the cost of increased adhesion; the resident trait $\hat{z}$ gradually declines and all individuals are eventually ungrouped. This population structure corresponds to permanently unicellular organisms. If initial adhesiveness is greater than $z^*$, instead, adhesiveness keeps increasing until its maximum value $\hat{z} = 1$ and all individuals belong to groups of identical size $T$. This outcome corresponds to the usual setting of multiplayer game-theoretical models.

The threshold $z^*$ increases with $T$ due to diminishing direct benefits, and converges to a maximum value

$$z^*_\infty = \frac{2c}{b} \qquad (4)$$

in the limit of infinite $T$. *Figure 3* displays the threshold $z^*$ as a function of the benefit-to-cost ratio $b/c$ for two different values of $T$ (blue and red full lines) and in the limit case $T \longrightarrow +\infty$ (black full

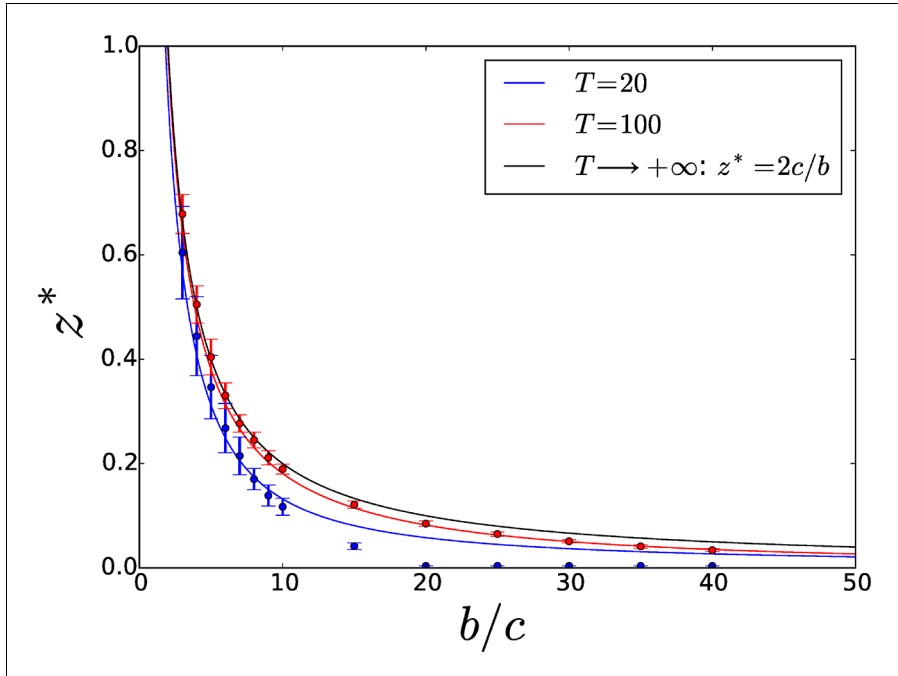

**Figure 3.** Threshold adhesiveness value $z^*$ required for the evolution of increased adhesion. In the case of group formation by attachment, the theoretical value of $z^*$ in the limit of infinite $T$ is $z^* = 2c/b$, as demonstrated in Section 4.1. Analytical thresholds (full lines) as well as numerical estimations (circles) are displayed for small (20) and large (100) values of $T$. As $T$ decreases, threshold values decrease too because of enhanced direct benefits. Numerical results are consistent with analytical predictions. Error bars indicate the variability—associated with the finite size of the population—in the estimation of the threshold across multiple computations of the aggregation process.

line). Overcoming adhesiveness level $z_\infty^*$ guarantees the selection of increased adhesiveness even in cases when groups are allowed to be very large, as commonly happens in microbial populations. The equilibrium $z^*$ can be classified as a 'garden of Eden', i.e. a noninvasible equilibrium that is not convergence-stable (*Nowak, 1990*) (see Section S2 of the SI).

In the following section, we focus on the evolutionary trajectories leading to an increase in adhesiveness, and discuss how the social nature of the trait changes in the course of evolution.

## 4.2 Social behavior along evolutionary trajectories

Cooperation, defined as a behavior that increases other individuals' fitness at a personal cost, has been classified in two categories, depending on the sign of the net effect of the behavior on the cooperator (*West et al., 2007*; *Wilson, 1975*, *1990*). Indeed, marginal gains retrieved from a cooperator's own contribution to the common good might be large enough to compensate its systematic cost: in this case, cooperation is termed *mutualistic* or *directly beneficial*, otherwise, it is coined *altruistic* (*West et al., 2007*). Note that in both cases, cooperators fare worse than the defective members of the same group. The distinction in the status of cooperative acts is also known as weak vs. strong altruism (*Wilson, 1975*, *1990*; *Fletcher and Doebeli, 2009*).

In randomly assorted populations, directly beneficial cooperation is favored by natural selection while altruistic cooperation is not (*Wilson, 1975*). Therefore, directly beneficial cooperation can be expected to establish first, as it is more easily obtained, potentially providing the substrate for altruistic forms of cooperation to spread later.

This idea seems to be supported by the common observation of directly beneficial behavior in microbes. In invertase-secreting yeast, a small proportion of the hydrolized glucose is retained by the producer, providing advantage to cooperator cells at low frequencies (*Gore et al., 2009*). Similarly, the bacterium *Lactococcus lactis* expresses an extracellular protease that helps transform milk

proteins into digestible peptides: *Bachmann et al. (2011)* showed that such cooperative behavior can persist owing to a small fraction of the peptides being immediately captured by the proteolytic cells. In *Pseudomonas aeruginosa* colonies grown on solid substrates, the diffusion of pyoverdine is locally confined as cells are densely packed, relaxing the burden of public good production (*Julou et al., 2013*). On the other hand, extremely sacrificial behavior such as altruistic suicide seems restricted to Myxobacteria and cellular slime moulds and its evolution appears to require mechanisms of reinforcement, though the peculiar life cycle of such organisms reduces the influence of genetic relatedness.

It is important to notice that the status of a cooperative trait (whether directly beneficial or altruistic) can be defined only relatively to a given population structure. In the case of linear PGGs within groups of fixed size $N$, direct benefits of a cooperator amount to $b/N$ and cooperation is mutualistic whenever $b/N > c$. Group size is thus crucial to define the status of a cooperative trait; the maintenance of cooperation in several models actually owes to alternating phases when it is altruistic or directly beneficial, generally because of group size variations (*Hauert et al., 2002*; *Fletcher and Zwick, 2004*; *Killingback et al., 2006*).

In our model, the population structure depends on the value of the social trait itself. At any point along an evolutionary trajectory, we can compute the average 'gain from switching' (*Peña et al., 2014*) of a more adhesive mutation, that quantifies the balance between the additional cost of increased adhesion, and the marginal return from its additional contribution to group cohesion. Here, the gain from switching is calculated in the social context established by less adhesive residents, characterized by the group size distribution $g(n, \hat{z}, \hat{z})$. We call 'altruisitic' any positive mutation on adhesiveness such that its gain from switching is negative, and 'directly beneficial' any mutation whose gain from switching is positive. These definitions extend those for groups of fixed size and encapsulate the concept of altruism as a situation in which individual costs are not immediately recovered by marginal benefits, all other things unchanged (but see *Kerr et al. (2004)* for other individual-based interpretations of altruism).

The condition for a small positive mutation to be altruistic in a resident population of trait $\hat{z}$ is, as showed in Section S3 of the SI:

$$r_{alt}(\hat{z})\, b < c \tag{5}$$

where:

$$r_{alt}(\hat{z}) = \sum_{n \geq 2} \frac{g(n, \hat{z}, \hat{z})}{n}. \tag{6}$$

This condition is comparable to that found in the classical PGG with fixed group size ($b < N\, c$), as $1/r_{alt}(\hat{z})$ is homogeneous to a group size and can be linked to the average group size in the population. Indeed, for the differential attachment model of Section 2.2,

$$r_{alt}(\hat{z}) = \frac{\hat{z}}{\hat{\gamma_{\hat{z}}}}, \tag{7}$$

where $\hat{\gamma_{\hat{z}}}$ is the average group size (across groups, not individuals—not to be confused with insider's view of group size or crowding (*Jarman, 1974*; *Reiczigel et al., 2008*)). In highly adhesive populations ($\hat{z} \approx 1$) with groups of size $T$ and no ungrouped individuals, *Equation 5* is then simply $b/c < T$; while in poorly adhesive populations ($\hat{z} \approx 0$) where all individuals are ungrouped, no direct benefit is possible, so that $r_{alt}$ vanishes and *Equation 5* is always satisfied.

*Figure 4* displays, for various values $\hat{z}$ of the resident trait, the benefit-to-cost ratio $1/r_{min}$ above which an increase in adhesiveness is favored and the benefit-to-cost ratio $1/r_{alt}$ below which such a mutation is classified as altruistic. Let us suppose that the benefit-to-cost ratio $b/c$ remains fixed all along an evolutionary trajectory as the resident trait varies. Depending on its values, an increase in adhesiveness can be attributed a different social status: (1) if $b/c > T$, social mutations are altruistic at the onset of social evolution, and directly beneficial when $\hat{z}$ becomes larger; (2) if $2 < b/c < T$, social mutations are altruistic throughout the evolutionary trajectory; (3) if $b/c < 2$, social mutations never invade.

The first case in particular demonstrates that the status of a social mutation can change along an evolutionary trajectory, and that this change is more likely to occur the smaller the maximal size

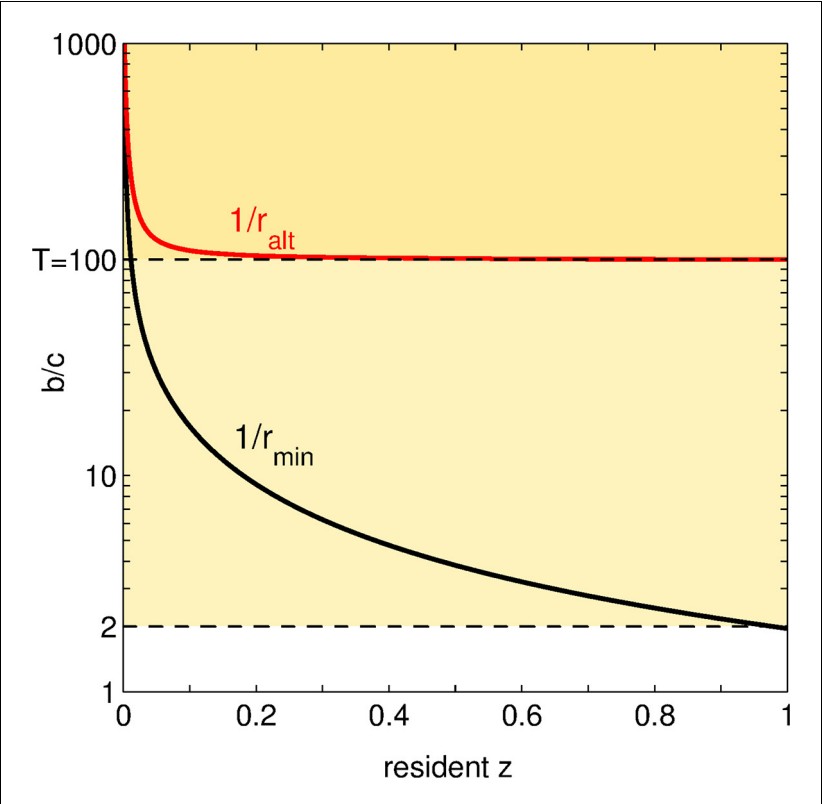

**Figure 4.** Status of social mutations. For any resident adhesiveness value $\hat{z}$ between 0 and 1, we display, in black: the minimal benefit-to-cost ratio $1/r_{min} = 2/\hat{z}$ for a social (or positive) mutation to be selected; in red: the maximal benefit-to-cost ratio such that this mutation is altruistic. Let us choose a fixed $b/c$ (i.e. an horizontal line in the graph). According to the value of $b/c$, the fate and the social status of positive mutations change. For low $b/c$ ($< 2$), all social mutations are altruistic but none of them is ever selected: the population is doomed to full asociality. For intermediate $b/c$ (between 2 and $T$), social mutations are favored as soon as $\hat{z}$ overcomes a threshold (crossing of the black line with the horizontal line $y = b/c$), and are altruistic all along the evolutionary dynamics. For large $b/c$ ($> T$), once the threshold is overcome and $\hat{z}$ increases, social mutations are altruistic until some value of $\hat{z}$ (crossing of the red line with the horizontal line $y = b/c$); afterwards, social mutations turn directly beneficial.

groups can attain. Contrary to what we expected, the social dilemma raised by costly mutations in adhesiveness relaxes over the course of evolution, leading from altruistic to mutually beneficial behavior. Indeed, even when the benefit-to-cost ratio is large (greater than the maximal group size $T$), social mutations are not directly beneficial at start. This is because, adhesiveness being small, chances are high to be left outside groups and miss any group-related benefit whatsoever.

This suggests that altruism may play a more important role in the origin of social behavior than it is currently assumed, and that it could be a first attainable step even when it involves large costs, eventually paving the way to the evolution of less sacrificial behavior.

## 4.3 Individual-based simulations

The generality of the analytical results obtained in Sections 3 and 4 rely on a number of hypotheses that are not strictly realized in actual biological populations: infinite population size; infinitesimal mutational steps; the equivalence between positive growth rate (when rare) and the fixation of the trait; the linear dependence of group-related benefits on group average composition. Whereas the soundness of adaptive dynamics equations as limits of individual-based processes has been mathematically proved (*Champagnat and Lambert, 2007*), the introduction of potential nonlinearities in the payoff function of our model quickly makes analytical calculations intractable.

We thus implemented the life cycle described in Section 2.2 in an individual-based model (detailed in Section 4 of the SI). This way, we could check the validity of the results of Section 4.1, here summarized by Pairwise Invasibility Plots (PIP) (*Geritz et al., 1998*), and repeat the analysis also for biologically interesting, non-linear payoff functions.

*Figure 5* displays the PIP for the individual-based model, representing the sign of the invasion fitness for each pair of resident $\hat{z}$ and mutant $z$ trait values. A single interior singular strategy $z^*$ can be observed and characterized as a 'garden of Eden' in accordance with the analytical results of Section 4.1: in a neighbourhood of $z^*$, the population can be successively invaded either by mutants with decreasing trait values until $\hat{z} = 0$, or by mutants with increasing trait values until $\hat{z} = 1$. *Figure 3* confirms the consistence between the thresholds obtained with the theoretical analyses (full lines) with those obtained by numerical simulations (circles) for two values of $T$. However, stochastic fluctuations due to a finite population size can, when the maximal group size $T$ is small, decrease such threshold to the extent that increased adhesiveness evolves for scratch (blue circles).

## 4.4 Evolutionary equilibria for other forms of collective fitness

While a linear PGG is a parsimonious—and tractable—way to depict social dilemmas occurring in multicellular aggregates, it is reasonable to imagine that actual cost and benefit functions are more convoluted. Notably, excessive group sizes and investments in adhesion might be constrained by intrinsic properties of microorganisms. By numerical simulations, we thus explored two alternative, nonlinear functional forms of the benefit or the cost function.

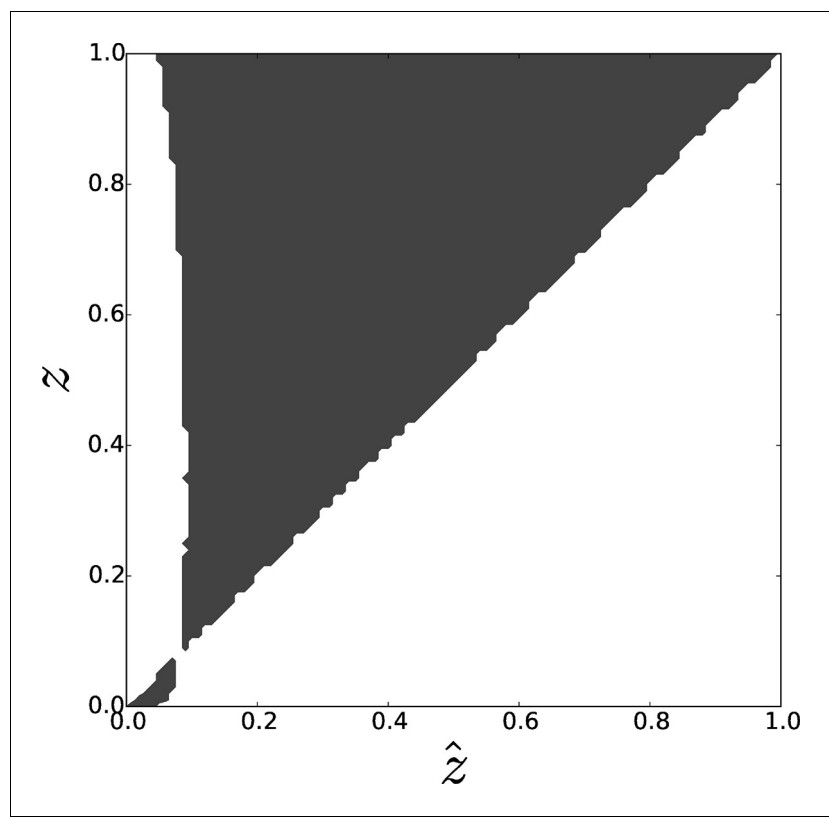

**Figure 5.** Pairwise invasibility plot obtained by simulation of the toy model for differential attachment. A positive invasion fitness (gray) means that the mutant can invade the population and replace the resident trait whereas a negative invasion fitness (white) means that the mutant is outcompeted. A singular point is found around $0.1 = 2c/b = z^*$, consistently with analytical predictions. This equilibrium can be characterized as a 'garden of Eden' (non-invasible repellor), which means that, depending on the position of the initial value $\hat{z}_0$ of $\hat{z}$ with respect to $z^*$, evolutionary dynamics leads to the selection of either $\hat{z} = 0$ (when $\hat{z}_0 < z^*$) or $\hat{z} = 1$ (when $\hat{z}_0 > z^*$), i.e. either full asociality or full sociality. Parameters: $T = 100$, $N = 5000$, $b/c = 20$.

First, we considered the situation when there is a physical upper limit to the size of functional groups. For instance, fruiting bodies of *Dictyostelium* become unstable and topple over if they are too large (*Savill and Hogeweg, 1997*). We model this by adding a group size threshold above which group benefits become null (*Figure 6*). Second, we modelled possible trade-offs between the production of the glue and other cell functions, so that producing large amounts of glue results in a disproportionate decrease in fitness (*Goymer et al., 2006*). This scenario corresponds to a cost function that is linear for small values of the adhesiveness $z$ but diverges as $z$ gets closer to $1$ (*Figure 7*).

Both cases lead to the appearance of a second internal singular strategy $z^+ < 1$ that can be characterized as a continuously stable (i.e. convergence-stable and noninvasible) strategy. This adaptive equilibrium is associated with populations structured in groups of sizes distributed around an average $z^+T$ and a proportion $1 - z^+$ of solitary individuals. Such an evolutionary outcome is more in line with observations of cellular slime moulds, where groups coexist with a fraction of nonaggregated cells (*Dubravcic et al., 2014*; *Tarnita et al., 2015*).

## 5 Discussion

The evolutionary stability of social behavior against the invasion by 'cheating' types, that do not contribute to collective functions, has been extensively studied within the formalism of evolutionary game theory. This paradox is typically modelled as a social dilemma involving two competing strategies: one 'social' (or 'cooperative') and one 'asocial' (or 'defective'). In this work, we model adhesiveness as a mechanism underpinning social behavior (*Garcia and De Monte, 2013*; *Garcia et al., 2014*; *Schluter et al., 2015*). Adhesiveness affects social interactions within groups, but also the formation of groups themselves. In the theoretical framework of adaptive dynamics, we describe the evolutionary trajectory of a continuous adhesion trait entailing a cost for its carrier. We address the very first steps in the emergence of collective behavior, as well as the long-term evolution of cohesive groups, in populations undergoing cycles of dispersal-aggregation-reproduction akin to those of facultatively multicellular microbes.

We assume that, other than its individual cost, adhesiveness provides collective benefits to groups by contributing to group cohesion. Benefits are equally shared between group members so

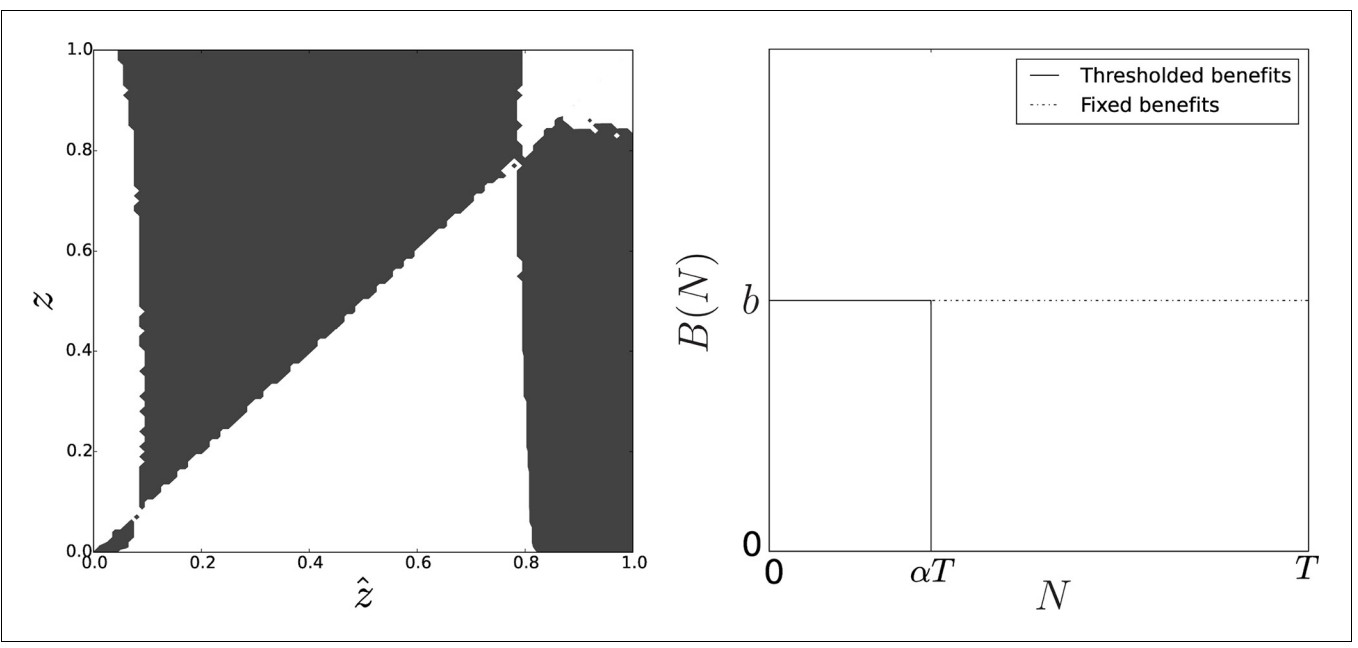

**Figure 6.** Pairwise invasibility plot obtained when group-related benefits are null above group size $\alpha T$. Here, the change in the benefit function leads to the appearance of a second equilibrium $z_+$ that is convergence-stable and non-invasible. As soon as the initial value of the adhesiveness trait $\hat{z}$ is larger than the adhesiveness threshold $z^*$, selection favors adhesion level $z^+$ at equilibrium. Parameters: $T = 100$, $N = 5000$.

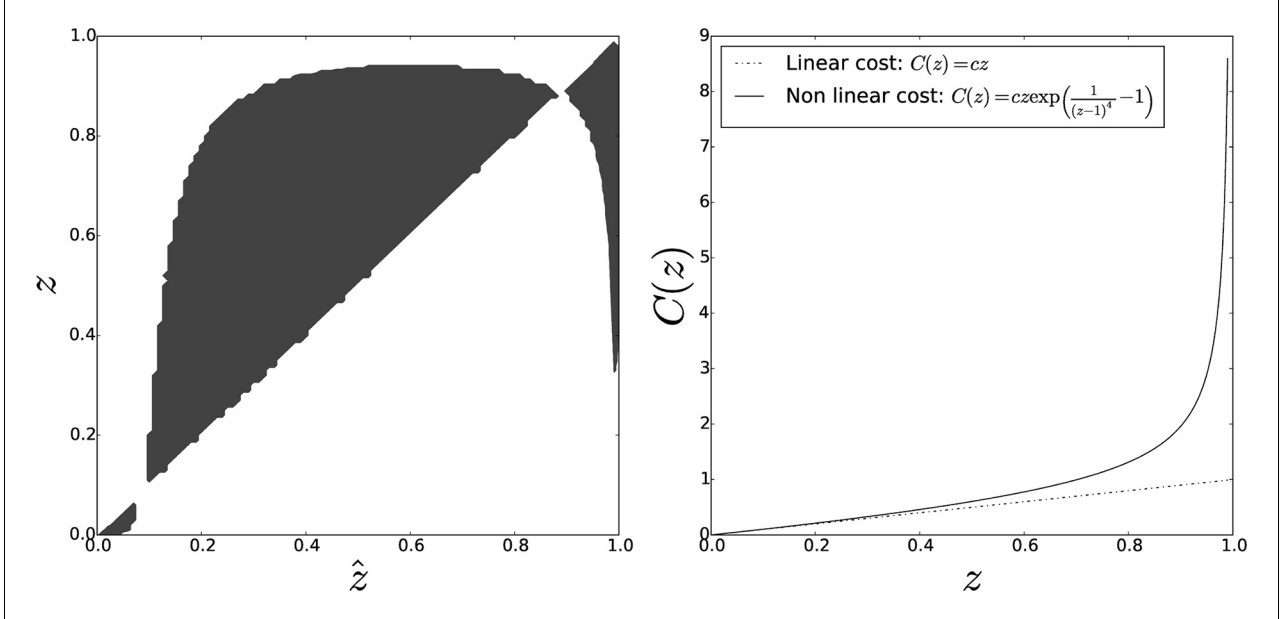

**Figure 7.** Pairwise invasibility plot obtained when individual cost diverges for large adhesiveness values. As in the previous case, an other equilibrium appears that is a CSS (convergence-stable strategy), hence the evolutionary endpoint as soon as the initial adhesiveness value overcomes the threshold $z^*$. Parameters: $T = 100$, $N = 5000$.

that individual gains depend only on group composition, and not group size. Even ahead of its effects on reproductive success, adhesiveness structures the population during the aggregation phase, as it increases the probability of attachment upon interaction. These assumptions model a feature often associated with social behavior in microbial populations: the production of extracellular glues. Glues are individually costly, since their production diverts energy allocated to growth. At the same time, they enhance cohesion and resilience of multicellular aggregates, and thus the fitness of the constituent cells.

For generic rules of group formation, we find that, if the initial value of the trait exceeds a value that is inversely proportional to the benefit-to-cost ratio, enhanced adhesiveness will be selected. In parallel to the rise in adhesiveness, the eco-evolutionary feedback between population composition and structure sustains the gradual surge of group size. This provides a scenario for the transition from poorly (or incidentally) social populations with small or loose groups to highly social populations structured in large cohesive groups. Unlike models with binary strategies, the evolution of sociality is possible from rare mutations in adhesiveness: the stickier type does not need to overcome a frequency threshold.

## 5.1 The ecology of adhesive cells

Social behavior is expected to occur in populations already endowed with some minimal ability of attachment before the multicellular stage evolved. Is this condition relevant to biologically realistic settings? Adhesiveness is a trait that is highly variable among microbial species, depending on the physical properties of the cell surface, as well as on the amount and quality of compounds they excrete. Directed selection experiments show that the secretion of extracellular products providing some kind of collective advantage—such as polysaccharides improving the strength of aggregates and stress resistance—can be achieved on a relatively fast time scale (*Rainey and Rainey, 2003*; *Xavier and Foster, 2007*). Overcoming a threshold in adhesiveness should thus not be a strong constraint on the evolution of social behavior, as long as the collective function provides a sufficient benefit.

Cells with no or little capacity of motion can rely on spatial patterning through clonal growth to ensure a high degree of local genetic and phenotypic homogeneity. Adhesion, on the other hand, can play such an assortative role for cells that actively move. Adhesiveness may therefore

significantly contribute to the evolution of multicellular organization by causing both assortment during group formation, and cell sorting at a later, developmental, stage (*Savill and Hogeweg, 1997*), thus underpinning both collective function and division of labor.

A change in adhesion per se, and not specifically to other cells, may however have drawbacks, if it is associated with a modification in dispersal properties. In *D. discoideum* at the onset of the aggregation phase, for instance, enhanced chemotaxis upon starvation (*Schäfer et al., 2013*) increases adherence to both the substrate and the neighboring cells. If higher adhesiveness drives more effective crawling, increased dispersal and the subsequent genetic heterogeneity in multicellular chimeras might oppose the selection for collective function. The emergence of sociality in spite of spatial mixing has been addressed in a simple model where cells are described as interacting persistent random walkers (*Garcia et al., 2014*). In this model, however, cell velocity and adhesiveness were independent parameters. Their concomitant variation might lead, in a spatially explicit setting, to insights on the conditions under which stable polymorphism is possible in a population with gradually evolving adhesiveness.

## 5.2 Solitary individuals

In the aggregation phase, populations of social amoebae *D. discoideum* do not only form multicellular groups, they also leave behind solitary cells. Recent experiments show that the existence of an ungrouped component is widespread in lab and wild strains. Theoretical arguments support the idea that nonaggregated cells play an essential role in the evolution of some peculiarities of those organisms (*Dubravcic et al., 2014*; *Tarnita et al., 2015*; *Rainey, 2015*). Neglecting cells that are outside groups alters population-level statistics, such as the average fitness of a given strategy, and thus affect the prediction of evolutionary outcomes.

Although widely disregarded in experiments, typically focused on the multicellular body, the possible role of solitary cells in the evolution of cooperative or social behavior has been a recurrent theme in theoretical works. Early simulations modeling the evolution of cellular slime moulds suggested that individuals 'diffusing' out of groups may contribute to the maintenance of cooperative behavior by forgoing the systematic cost of grouping, together with its potential benefits (*Armstrong, 1984*).

*Hauert et al. (2002)* stressed that a 'loner' strategy—whereby individuals adopt an autarkic lifestyle—can provide a way out of the deadlock of the tragedy of the commons by a periodic soaring of individuals that opt out of the collective enterprise. Even though the predicted oscillations in group size have never, to our knowledge, been observed in microbial populations, they might be conceived as primitive life cycles including a collective phase.

Solitary behavior has also been explained as an adaptation to variable environmental conditions (*Dubravcic et al., 2014*; *Tarnita et al., 2015*; *Rainey, 2015*): cells hedge their evolutionary bets between commitment to a developmental program with uncertain outcome (they may end up in the stalk or in the spores) and the wait for the re-establishment of conditions favorable to growth. The theoretical prediction in this case is that the fraction of lonely cells reflects the optimal probability of remaining outside groups given the pattern of environmental variation.

The evolutionary model discussed here also supports the claim that lonely cells, which happen to remain outside groups by chance, are of primary importance for the onset of sociality. As discussed in Section 3 for a general group formation process, and illustrated in Section 4 with a toy model for aggregation, nonaggregated individuals might tip the balance in favor of stickier types. This conclusion is reinforced by the fact that, in our model, the fitness of an individual depends on the composition of the group it belongs to—as opposed to the assumption in *Dubravcic et al. (2014)* and *Tarnita et al. (2015)* that, when in groups, the probability of becoming spores is constant. At the evolutionary equilibrium, however, the ungrouped component of the population vanishes, unless nonlinearities are introduced in the functional dependence of collective benefits and/or individual costs on adhesiveness, as observed in Section 4.3.

Some important differences between *Dubravcic et al. (2014)* and *Tarnita et al. (2015)* and our model should be noted. When group formation is based on adhesiveness, lonely individuals occur also in the absence of temporal variation of the environment and of selective advantage due to dispersal. Moreover, in our model, being ungrouped is not a strategy determined by a fixed probability, as failure to join a group is contingent on the emergent population structure. A pivotal issue is thus when the decision to join groups is made by microbial organisms. Populations of facultatively

multicellular species face two 'lotteries' where stochasticity affects the fitness of an individual: joining groups and, once in a group, reproducing or dying. If the trait under selection is, as in *Dubravcic et al. (2014)* and *Tarnita et al. (2015)*, the probability of staying alone, cells decide to be in a group before the fitness cost of giving up reproduction is possibly assigned. If it is adhesiveness, as in our model, social behavior in groups is decided at the same time as aggregative behavior. Experimental evidence suggests that conditions preceding the aggregation phase influence the partition between lonely and aggregated cells (*Dubravcic et al., 2014*), and between stalk and spore cells (*Gomer and Firtel, 1987*; *Jang and Gomer, 2011*). The role of adhesiveness itself in influencing such decisions is unknown. Further experiments are needed, together with models that explore the possible role of competition within aggregates, to clarify to what extent social interactions within the multicellular phase are linked to group size distribution, including singletons.

### 5.3 The nature of social traits

Directly beneficial cooperation is by definition more easily established than altruistic cooperation (*West et al., 2007*). Example of microbial populations where individuals draw direct benefits from the production of extracellular compounds are numerous (*Gore et al., 2009*; *Bachmann et al., 2011*; *Julou et al., 2013*; *Zhang and Rainey, 2013*). One could thus expect that, by sustaining the onset of cooperation, directly beneficial behavior sets the scene for the later establishment of altruistic traits, of which 'suicide for the good of the group' is emblematic.

Deciding the nature (altruistic or not) of a given social mutation is not just a matter of costs and benefits, but also depends on the way the population is structured. As a consequence, the nature of a directional change in the trait value can vary along an evolutionary trajectory, if it entails modifications in the average local interaction environment. We show that the status of mutations increasing adhesiveness can change along with the gradual evolution of the trait, even when benefit and cost parameters are fixed. When adhesion starts to increase, mutations are always altruistic, meaning that they entail a net cost to the individual compared to the resident trait. Nonetheless, these altruistic mutants thrive because of their role in the earlier aggregation phase. For large enough benefit-to-cost ratios, though, positive mutations become directly beneficial at a later time along the evolutionary trajectory. Altruistic behavior emerges first, and direct benefits only appear afterwards. This means that directly beneficial interactions, while frequently observed in nature, might not necessarily be the stepping stone for the evolution of cooperation, but rather an evolutionary endpoint.

The gradual increase in adhesiveness is a mechanistic—and seemingly readily achievable—way by which unicellular organisms may have made the transition to complex, multicellular organization (*De Monte and Rainey, 2014*; *Rainey and De Monte, 2014*). Such a mechanism not only accounts for assortment, but potentially underpins spatial segregation in tissues (*Steinberg, 2007*; *Marée and Hogeweg, 2001*), through which evolutionary forces might have moulded the developmental program of the multicellular body. The relevance of adhesion in structuring the collective phase of organisms that aggregate from sparse cells however requires more experimental observations, that will hopefully shed light on the biological and physical processes involved in the major evolutionary transition to multicellularity.

## Acknowledgements

The authors are very grateful to Paul Rainey, Corina Tarnita, three anonymous referees and the scientific editor for providing critical comments and useful suggestions to improve the manuscript. TG was supported by the UPMC/IRD program 'Modélisation des Systèmes Complexes'. SDM acknowledges support through the project IAMEE of the Paris Sciences et Lettres program 'Structuration de la Recherche'.

## Additional information

### Funding

No external funding was received for this work.

## Author contributions

TG, Conceived the model; analysed the model; wrote the article; GD, Analysed the model; did the computations; wrote the article; SDM, Conceived the model; wrote the article

## Author ORCIDs

Guilhem Doulcier, http://orcid.org/0000-0003-3720-9089

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

## Appendix A1: Selection gradient for a generic aggregation process

Let us consider a monomorphic population of resident $\hat{z}$-individuals, among which an infinitely small fraction of mutants with trait $z = \hat{z} + dz$ arises. As mutants are rare, the average resident individual belongs to a group composed of $\hat{z}$ individuals, so that the mean level of sociality in its group is simply $\bar{z} = \hat{z}$. Its payoff in a PGG thus writes

$$P_n(\hat{z}, \hat{z}) = b\hat{z} - c\hat{z} \tag{A1}$$

and is independent of group size.

Similarly, mutants are considered to be rare enough so that the probability of two mutants belonging to the same group is negligible. The average level of sociality experienced by a mutant in a group of size $n$ (that is, with $n - 1$ individuals carrying the trait $\hat{z}$) is then $\bar{z} = (z + (n-1)\hat{z})/n = \hat{z} + (z - \hat{z})/n$. The mutant payoff within a group of size $n$ hence writes

$$P_n(z, \hat{z}) = b\hat{z} + b\frac{z - \hat{z}}{n} - cz \tag{A2}$$

The resident population structure is described by two distributions. The group-size distribution $g(n, z, \hat{z})$ describes the probability that an individual carrying trait $z$ in a population composed of individuals of trait $\hat{z}$ is found, when the aggregation process is over, in groups of size $n > 1$. The distribution $u(\hat{z}, \hat{z})$ (resp. $u(z, \hat{z})$) is the fraction of $\hat{z}$-individuals (resp. $z$-individuals) left ungrouped. Weighting for all possible group sizes, the final net payoff of the average resident is:

$$P(\hat{z}, \hat{z}) = \sum_{n \geq 2} g(n, \hat{z}, \hat{z}) \; P_n(\hat{z}, \hat{z}) - c\hat{z} = b\hat{z}[1 - u(\hat{z}, \hat{z})] - c\hat{z} \tag{A3}$$

and, for a mutant,

$$\begin{aligned}
P(z, \hat{z}) &= \sum_{n \geq 2} g(n, z, \hat{z}) P_n(z, \hat{z}) - cz \\
&= b\hat{z}[1 - u(z, \hat{z})] + b(z - \hat{z}) \sum_{n \geq 2} \frac{g(n, z, \hat{z})}{n} - cz
\end{aligned} \tag{A4}$$

We can calculate the invasion fitness of mutants:

$$\begin{aligned}
S(z, \hat{z}) &= P(z, \hat{z}) - P(\hat{z}, \hat{z}) \\
&= b\hat{z}[u(\hat{z}, \hat{z}) - u(z, \hat{z})] + b(z - \hat{z}) \sum_{n \geq 2} \frac{g(n, z, \hat{z})}{n} - c(z - \hat{z})
\end{aligned} \tag{A5}$$

Supposing that all functions in **Equation A1** Appendix A1: Selection gradient for a generic aggregation process are differentiable, we can easily derive the selection gradient:

$$\left. \frac{\partial S(z, \hat{z})}{\partial z} \right|_{z=\hat{z}} = \lim_{z \longrightarrow \hat{z}} \frac{S(z, \hat{z})}{z - \hat{z}} = \lim_{z \longrightarrow \hat{z}} \left( -b\hat{z} \frac{u(z, \hat{z}) - u(\hat{z}, \hat{z})}{z - \hat{z}} + b \sum_{n \geq 2} \frac{g(n, z, \hat{z})}{n} - c \right) \tag{A6}$$

The sum being bounded, the limit and the sum can be exchanged. The selection gradient is thus:

$$\frac{\partial S(z,\hat{z})}{\partial z}\bigg|_{z=\hat{z}} = -b\hat{z}\frac{\partial u(z,\hat{z})}{\partial z}\bigg|_{z=\hat{z}} + b\sum_{n\geq 2}\frac{1}{n}g(n,\hat{z},\hat{z}) - c \tag{A7}$$

# Appendix A2: Application to aggregation by differential attachment

In this Appendix, we compute the terms in equation *Equation A1* for the model of aggregation by attachment, introduced in Section 3.

In a population of resident trait $\hat{z}$, a rare mutant has a negligible probability to be a recruiter. It can therefore either remain alone with probability

$$u(z,\hat{z}) = 1 - \sqrt{z\hat{z}} \tag{A8}$$

or belong to a group of size $n \geq 2$ with probability:

$$g(n,z,\hat{z}) = \sqrt{z\hat{z}} \binom{T-2}{n-2} \hat{z}^{n-2} (1-\hat{z})^{T-n} \tag{A9}$$

Indeed, dyadic interactions between the mutant and a resident succeed with probability $\sqrt{z\hat{z}}$. If they fail (with probability $1 - \sqrt{z\hat{z}}$), the mutant remains alone. The mutant will instead belong to a group of size $n \geq 2$ if it attaches to its recruiter (with probability $\sqrt{z\hat{z}}$), and this in turn must recruit $n-2$ resident individuals among the $T-2$ remaining (each with probability $\sqrt{\hat{z}\hat{z}} = \hat{z}$).

We can now compute

$$h(\hat{z}) = -\frac{\partial u(z,\hat{z})}{\partial z}\big|_{z=\hat{z}} = \frac{1}{2}. \tag{A10}$$

The sum $b\sum_{n\geq 2} \frac{1}{n} g(n,\hat{z},\hat{z})$ depends on $T$, and can be calculated using the following general formula:

$$\sum_{k=0}^{n} \frac{1}{k+2} \binom{N}{k} A^k B^{N-k} = \frac{(N+2)A(A+B)^{N+1} - (A+B)^{N+2} + B^{N+2}}{A^2(N+1)(N+2)} \tag{A11}$$

yielding:

$$\sum_{n=2}^{T} \frac{1}{n} g(n,\hat{z},\hat{z}) = \frac{T\hat{z} - 1 + (1-\hat{z})^T}{\hat{z}(T-1)T} \tag{A12}$$

which is positive and converges toward $0$ when $T \longrightarrow +\infty$. Therefore, $z_\infty^* = 2c/b$ (see main text) is the trait value above which higher adhesiveness will spread with certainty. Figure 3 of the main text displays the threshold value $z^*$ for both finite and infinite $T$.

## Characterization of the singular point

The singular point $z^* = 2c/b$ can be fully characterized calculating the second derivatives of the invasion fitness $S_{\hat{z}}(z)$ (*Geritz et al., 1997*; *Waxman and Gavrilets, 2005*). Let us denote

$$d_{11} = \frac{\partial^2 S_{\hat{z}}(z)}{\partial \hat{z}^2}\big|_{z=\hat{z}=z^*} \tag{A13}$$

and

$$d_{22} = \frac{\partial^2 S_{\hat{z}}(z)}{\partial z^2}\Big|_{z=\hat{z}=z^*}, \tag{A14}$$

then the properties (invasibility and convergence-stability) of the equilibrium are provided by the signs of resp. $d_{22}$ and $d_{11} - d_{22}$.

Using the expression of $g(n, z, \hat{z})$ in **Equation A9** and the formula in **Equation A11**, we find, in the limit $T \longrightarrow +\infty$ (details not shown):

$$d_{11} = -\frac{3b}{2} - \frac{b^2}{8c} \text{ and } d_{22} = -\frac{b^2}{8c} \tag{A15}$$

so that $d_{22} < 0$ and $d_{11} - d_{22} < 0$. The equilibrium is then a 'garden of Eden' (i.e. a noninvasible strategy that is not convergence-stable).

# Appendix A3: Condition for altruistic sociality

The group size distribution after aggregation in an $\hat{z}$-resident population is given by $g(n, \hat{z}, \hat{z})$ with $n > 1$. A more adhesive mutant of trait $z > \hat{z}$ appearing in this population structure would have an average net payoff:

$$b \sum_{n \geq 2} g(n, \hat{z}, \hat{z}) \frac{z + (n-1)\hat{z}}{n} - cz \tag{A16}$$

while the less adhesive resident gets:

$$b \sum_{n \geq 2} g(n, \hat{z}, \hat{z}) \hat{z} - c\hat{z} \tag{A17}$$

Therefore, the mutant will be classified as altruistic if:

$$b \sum_{n \geq 2} g(n, \hat{z}, \hat{z}) \frac{z + (n-1)\hat{z}}{n} - cz < b \sum_{n \geq 2} g(n, \hat{z}, \hat{z}) \hat{z} - c\hat{z} \tag{A18}$$

i.e. whenever:

$$b \sum_{n \geq 2} g(n, \hat{z}, \hat{z}) \frac{z - \hat{z}}{n} < c(z - \hat{z})$$

$$\Leftrightarrow \frac{c}{b} > \sum_{n \geq 2} \frac{g(n, \hat{z}, \hat{z})}{n} \quad := r_{alt}(\hat{z}). \tag{A19}$$

The threshold value $1/r_{alt}(\hat{z})$ for $b/c$ that separates altruistic from directly beneficial behaviour is homogeneous to a group size, and can be linked to the average size of groups in the population (outsider's view of group size, i.e. averaged across groups and not individuals). Indeed, excluding ungrouped individuals, the mean group size is given by:

$$\hat{\gamma}_{\hat{z}} = \frac{\sum_{n \geq 2}^{T} n f(n)}{\sum_{n \geq 2}^{T} f(n)} \tag{A20}$$

where $f(n) = \frac{g(n, \hat{z}, \hat{z})/n}{\sum_{k \geq 2}^{T} g(k, \hat{z}, \hat{z})/k}$ is the frequency of $n$-sized groups over all the population. Thus,

$$\hat{\gamma}_{\hat{z}} = \frac{\sum_{n \geq 2}^{T} g(n, \hat{z}, \hat{z})}{\sum_{n \geq 2}^{T} \frac{g(n, \hat{z}, \hat{z})}{n}} = \frac{1 - u(\hat{z}, \hat{z})}{r_{alt}(\hat{z})} \tag{A21}$$

As the proportion of ungrouped individuals $u(\hat{z}, \hat{z}) = 1 - \hat{z}$, we finally get:

$$r_{alt}(\hat{z}) = \frac{\hat{z}}{\hat{\gamma}_{\hat{z}}} \tag{A22}$$

## Appendix A4: Individual-based simulations

The model for aggregation by attachment explained in section 2.2 is numerically simulated by an individual-based model, that allows to explore the evolution of adhesiveness in general settings, including finite-size populations, and nonlinear payoff functions. Our implementation is freely available at:

https://github.com/geeklhem/pimad

and can easily be extended to other group formation processes or functional forms of fitness.

Each player in a population of size $N$ (multiple of $T$) is characterized by two variables: its phenotype (which can take two values: resident or mutant) and aggregation status (ungrouped/grouped).

At the beginning of the life cycle, the aggregation status is set to ungrouped. The first variable is randomly assigned to mutant with a probability $ip$. For the sake of reducing computation time, we start with a small initial proportion of mutants, instead of implementing explicitly a mutation-substitution process. We tested in a few cases that the two processes were equivalent to determine the invasion fitness (a quantity independent on the time to reach fixation of the mutant).

During the Aggregation Phase (AP), players are randomly distributed among patches of size $T$. Then an individual is randomly selected to become the 'recruiter'. Finally we set the aggregation status of other players to grouped with a probability equal to the geometric mean of their own and their recruiter's phenotypic values (see section 2.2).

The Reproductive Phase (RP) consists in first computing the payoff of each individual in the population, and then letting them reproduce accordingly. First, we attribute a payoff to each individual according to the PGG played in its patch: each player pays a cost depending on its trait value and grouped players receive a benefit that depends on the average trait value within their group. For a linear PGG, the benefit $B$ and the cost $C$ are defined as in the main text: $B(\hat{z}) = B\hat{z}$ and $C(z) = cz$. We then simulate a birth-death process by duplicating each player with a probability equal to its payoff linearly normalized across the population to fall between $0$ and $1$. The offspring inherits the phenotype of its parent. Finally, we randomly remove from the previous population a number of individuals equal to the total number of offsprings, so that population size remains constant.

During the Dispersal Phase (DP), we reinitialize the aggregation variable of each player to ungrouped and shuffle the population.

## Numerical estimation of the invasion fitness

The intrinsic stochasticity of the RP's birth-death process is obvious when one observes the spread of a rare mutant. Indeed, in many cases the mutant goes extinct just because of demographic fluctuations.

The empirical invasion fitness $s$ has thus to be computed as an average over many realizations. We estimate it numerically by a maximum likelihood method, using a protocol inspired by **Chelo (2014)**: we simulate $k$ trajectories and call $n_{10}^{(i)}$ $(i = 1 \ldots k)$ the number of mutants at generation 10 of the $i$-th realization. The log-likelihood of $s$ given the data is:

$$L(s) = \sum_{i=1}^{r} \log\left[\Pr\left(n_{10}^{(i)} | s, n_0, N\right)\right] \tag{A23}$$

where $n_0$ is the initial number of mutants, $N$ the population size and

$$\Pr\left(n_{10}^{(i)}|s, n_0, N\right) \qquad \text{(A24)}$$

is the probability of having $n_{10}^{(i)}$ mutants in the 10<sup>th</sup> generation. The latter probability is computed using a selection-drift model: the number of mutants in generation $n_{t+1}$ is drawn from a binomial law with parameters $\left(N, \frac{n_t s}{N + n_t(s-1)}\right)$. For a given set of parameter values ($s$, $n_0$, $N$), $P$ follows a normal distribution with statistics assigned based on $k = 1000$ numerical realizations of $n_{10}$. The maximum likelihood estimator of the invasion fitness is obtained by taking the value of $s$ for which $L(s)$ is the highest:

$$\hat{s} = \arg\,\max_{s}\,L(s) \qquad \text{(25)}$$

Adaptive dynamics assumes that the mutant invades (i.e. the invasion fitness is positive) if $\hat{s} > 1$. Longer simulations performed for several close values of the resident and mutant trait confirmed that a positive invasion fitness indeed leads to a complete replacement of the resident trait by the mutant trait. Coexistence of mutant and resident traits over long timescales was never observed.

