## [Decision Letter]

Thank you for submitting your work entitled "The evolution of adhesiveness as a social adaptation" for peer review at *eLife*. Your submission has been favorably evaluated by Diethard Tautz (Senior editor), a Reviewing editor, and three reviewers.

The reviewers have discussed the reviews with one another and the Reviewing editor has drafted this decision to help you prepare a revised submission.

General assessment and major comments:

The Garcia et al. manuscript is a very elegant take on the evolution of social traits in the absence of explicit spatial structure or reciprocity. The paper argues that cell adhesiveness is a likely first step in the major evolutionary transition from unicellular organisms to multi-cellular organisms. However, since producing adhesives is costly, the usual conflict between individual and group level benefits occur.

This paper explores the origins of cellular adhesiveness as both a means of aggregating individuals into groups (creating population structure) and as a trait with fitness consequences to individuals and groups (once groups are formed). The analytical version of the model is based on adaptive dynamics and therefore allows for the exploration of small changes to a continuous trait. The main findings are that adhesiveness can evolve to high levels via small steps and that this can be true even when the trait initially has a net cost (strong altruism), as opposed to weak altruism paving the way for strong altruism. Overall, the article seems very well written and interesting and could make an important contribution.

The paper is full of good ideas and interesting insights on the origins of multicellularity, but all referees express concerns in terms of the presentation:

1) The model is not described very well. It is difficult to understand how the simulation model worked. It does not seem to be possible, based on the material in the paper, to reproduce the figures, for example:

What is N? Is it a multiple of T? Why not have N=T? What is the vertical axis in Figure 4?

How does the RP work? Is it a Moran process? The first paragraph of subheading “Numerical simulations” is hard to follow. What is the "competition between groups"? If each individual leaves at most one offspring, how is the population kept fixed?

2) The authors expand on their own work to show that adhesiveness (the trait that makes individuals more likely to attach to others and form groups) can evolve in a population of well-mixed individuals that keep aggregating and dispersing as part of their life cycle. This is potentially an important contribution to the study of social behavior because it would show that pre-existing assorting forces are not necessary for sociality to evolve. However, in paragraph three, Discussion, one of the main conclusions of this study – where adhesiveness is a continuous trait – is that the evolution of sociality is possible from rare mutations. However, at the same time, they can't really evolve from scratch. There needs to be a threshold amount of adhesiveness present in the resident type for a more sticky one to evolve, as discussed in the subheading “The ecology of adhesive cells” discussing why this is likely but I didn't find that discussion sufficiently persuasive. If the model shows that basically stickiness can't evolve from scratch, then how would there be a certain amount of stickiness present in the population to begin with? What are some examples where we expect this to have happened?

The authors highlight the adaptive dynamics (AD) analysis. It is described before the "real" model is described, and the simulation is described simply as a test of the "robustness" of the AD analysis. This seems backwards. The result of the AD analysis is not particularly interesting (limits at z=0 or z=1 depending on the starting point) and if AD was really the proper model then (as is pointed out many times in the paper) the process of increasing adhesiveness could never get started, due to the positive threshold that must be overcome. It seems the AD analysis is basically an interesting "sanity check" on the simulation model, that is valid in certain limiting cases. (The work of Champagnat in the reference, and elsewhere, shows just how "hard" it is to get an AD limit from a sequence of stochastic models.) This is not to make light of the nice analytical results in section 2, but we think it should be made clearer that the most important insights in the paper come from the computer simulation of the mathematically intractable versions of the model. (Assuming the authors agree with this.)

3) It is great that the authors talk about adhesiveness as a continuous trait, especially because this turns the very binary problem of cooperation-defection or social-asocial into a much more nuanced question. But despite posing the problem so generally, the authors still use the social-asocial language, even when it doesn't make much sense. For example in paragraph five, subheading “Social behavior along evolutionary trajectories: altruistic sociality”, where both the resident and the mutant are actually sticky but one is a tiny bit less so and is then called asocial. That seems confusing (despite the fact that many people do this) and we would suggest finding a clearer way to refer to these traits that fits with the continuum perspective which, again, I think is a very good one.

4) The difference between cooperation and altruism has always been confusing. Indeed the way the authors present it is one way in which it has been described but not the only one and this part of the manuscript and its conclusions can lead to confusion. It seems that a lot of what people have termed cooperation and studied in the context of the Prisoner's dilemma is altruism according to the author's definition (see e.g. Nowak et al. work). And in many works altruism has been defined as an extreme form of cooperation, such as suicide (stalk cells in *D. discoideum* or worker ants in eusocial colonies). The authors don't seem to be talking about the latter when they're saying that altruism can evolve first. They seem to be talking about the former, which most people would interpret as actual cooperation (as opposed to mutually beneficial behavior). We are perfectly aware that the semantics of this field is a mess, but one ought to be very careful in phrasing the conclusions of this part of the study. Perhaps moving away from cooperation-altruism and phrasing it more generally and in more nuanced ways would help. Another possibility, if the authors want to stick with cooperation-altruism, is to explain the different definitions and cite other literatures as well. Because the results of this section seem very similar to, for example, people claiming that cooperation (rather than altruism) can evolve on networks if b/c > k.

5) There is a related philosophical quibble concerning the distinction between net direct benefits (sometimes called weak altruism) and net costs (sometimes called strong altruism). If a social behavior gives a benefit to the actor that is greater than its cost, this may seem to be substantially different than the case where the actor's cost is greater than any benefit it receives. But it is relative fitness that matters and if the actor's social behavior benefits all group members, but others do not pay the cost borne by the actor, then there is no substantial difference. For simplicity assume that the actor provides benefit B to the group at cost C to the actor. If this benefit is split evenly among group members (including the actor) then actors (weak altruists) get B/n - C whereas other group members get B/n. Whether B/n - C is positive or negative doesn't affect the fact that actors in a group are at a disadvantage compared to non-actors in the group. Whether actors (altruists) are favored depends on population structure (assortment), not whether B/n - C is positive or negative. Weak altruists can be selected against if the interaction structure is unfavorable and strong altruists can be selected for if it is favorable. Of course the degree of assortment needed can be affected by the values of B and C (and therefore by whether B/n -C is positive or negative), but this is a matter of degrees of assortment, not a fundamentally different mechanism. Comparing two scenarios where B/n - C values are 2 and 6 respectively is not inherently different from comparing two scenarios where the values are -2 and +2 respectively. So it is not surprising or "counter-intuitive" that strong altruism can proceed weak altruism in the evolutionary trajectories of the model. The amount of assortment simply favors each in its turn.

6) As for the conclusion that the status of a social mutation can change along an evolutionary trajectory, that's a great point and an interesting conclusion. However, the conditions under which it happens seem very restrictive (the b/c ratio has to be very high). The authors should discuss the intuition and biological feasibility of such a scenario.

[Editors' note: further revisions were requested prior to acceptance, as described below.]

Thank you for resubmitting your work entitled "The evolution of adhesiveness as a social adaptation" for further consideration at *eLife*. Your revised article has been favorably evaluated by Diethard Tautz (Senior editor), a Reviewing editor, and three reviewers. The manuscript has been improved but there are some minimal remaining issues that would ideally be addressed before publication (as you will see, the only reason that the paper was not immediately accepted is to give you a chance to incorporate a reply to two queries), as outlined below:

Reviewer #1:

I think the authors have addressed all my concerns, so at this point I am ready to recommend the paper be published in *eLife*.

In my first review I made a comment that I would like to retract now. At the time I saw the AD analysis as being "uninteresting" because the result was simply a limit at z = 0 or 1 depending on the starting point. Soon after I posted that review I realized that this was not a reasonable complaint. Fortunately this comment was seen for what it was and no harm was done!

Reviewer #2:

I'm satisfied with the authors' revisions (and with the way they've addressed my concerns). I find the new manuscript acceptable.

Reviewer #3:

I think this revised manuscript is substantially improved and I am satisfied that the authors have addressed my comments from the initial submission. I recommend acceptance.

That said, I have a couple of additional comments:

In reading the response to reviewers on item 6 I was struck by the detail that in calculating the benefits and costs across the population, that all individuals (including loners) paid the cost of adhesiveness, while only those that made it into groups experienced the benefits. (I missed that detail in the first version, and its role in determining b/c ratios.) I don't think I understand the justification for this. If an extracellular matrix was the trait that caused adhesiveness and also benefited other group members, then loners might get at least some benefit from their own excretions of extracellular matrix. Also, one could imagine a different mechanism of adhesion that did not manifest itself unless a group was joined. In this case loners would not pay the cost. I would leave it up to the authors to decide whether providing some intuition behind this analysis decision is helpful.

Lastly, in paragraph one, Introduction, the problem of free riders is discussed. I'm wondering if that applies to this model? Is it true that one of the mechanisms that makes this model work is that free riding is mitigated because those that are less helpful automatically are more likely to be excluded from groups (they are less adhesive)? Again, I leave it up to the authors, but if having a single trait for both adhesiveness and group beneficial behavior is instrumental in reducing free-riding (and therefore instrumental in the reported results), it would be helpful to state this explicitly.

---

## [Author Response]

*1) The model is not described very well. It is difficult to understand how the simulation model worked. It does not seem to be possible, based on the material in the paper, to reproduce the figures, for example: What is N? Is it a multiple of T? Why not have N=T? What is the vertical axis in Figure 4? How does the RP work? Is it a Moran process? The first paragraph of subheading “Numerical simulations” is hard to follow. What is the "competition between groups"? If each individual leaves at most one offspring, how is the population kept fixed?*

We have added the subheading “Group formation based on attachment”, that describes the model for group formation based on attachment, extending the model introduced in Garcia & De Monte 2013 to the case of a continuous adhesion trait. This should clarify the role of the parameter T (in an infinite population). The algorithm used to create Figure 4, Figure 5 and Figure 6 is now detailed in subheading “Appendix A4: Individual-based simulations”. There, we explain that N (multiple of T) is the size of the finite simulated population, and how birth, death and dispersal are implemented.

*2) The authors expand on their own work to show that adhesiveness (the trait that makes individuals more likely to attach to others and form groups) can evolve in a population of well-mixed individuals that keep aggregating and dispersing as part of their life cycle. This is potentially an important contribution to the study of social behavior because it would show that pre-existing assorting forces are not necessary for sociality to evolve. However, in paragraph three, Discussion, one of the main conclusions of this study – where adhesiveness is a continuous trait – is that the evolution of sociality is possible from rare mutations. However, at the same time, they can't really evolve from scratch. There needs to be a threshold amount of adhesiveness present in the resident type for a more sticky one to evolve, as discussed in the subheading “The ecology of adhesive cells” discussing why this is likely but I didn't find that discussion sufficiently persuasive. If the model shows that basically stickiness can't evolve from scratch, then how would there be a certain amount of stickiness present in the population to begin with? What are some examples where we expect this to have happened?*

In subheading “Group formation based on attachment”, we have stressed the difference between a threshold in the frequency of the individuals carrying a trait, and a threshold in the value of the trait. The point we want to make is that, in most cases when the evolution of cooperation is considered 'paradoxical', models inevitably require that cooperators start from a minimal frequency in the population, sufficient for them to gather some advantage from the collective enterprise. Situations when cooperators are advantaged even when they are extremely rare typically rely on directly beneficial cooperation, in which the evolution of social behavior is intrinsically less challenging. In our model, more adhesive variants can evolve even when infinitesimally rare at the beginning, because they exploit the population structure set by the resident.

On the other hand, for such a structure to be able to ratchet adhesiveness up, the value of the trait must not be too low. If it is too low, the assortment provided by group formation is insufficient to support the success of increased adhesiveness. A condition like the former, resting on the initial frequency of the more adhesive type (and not on the initial value of the trait) would burden the selection of each successive positive mutation. Instead, the latter condition on the initial trait value applies only at the beginning of the evolutionary trajectory.

In subheading “The ecology of adhesive cells”, we explain why we find plausible that a small level of adhesiveness could have evolved independently from any social function. Indeed, extracellular polysaccharides are commonly produced as a means of protection from stressants. Moreover, in order to crawl, cells need to adhere to the substrate, which in some cases implies adhering to other cells as well.

*The authors highlight the adaptive dynamics (AD) analysis. It is described before the "real" model is described, and the simulation is described simply as a test of the 'robustness' of the AD analysis. This seems backwards. The result of the AD analysis is not particularly interesting (limits at z=0 or z=1 depending on the starting point) and if AD was really the proper model then (as is pointed out many times in the paper) the process of increasing adhesiveness could never get started, due to the positive threshold that must be overcome. It seems the AD analysis is basically an interesting "sanity check" on the simulation model, that is valid in certain limiting cases. (The work of Champagnat in the reference, and elsewhere, shows just how "hard" it is to get an AD limit from a sequence of stochastic models.) This is not to make light of the nice analytical results in section 2, but we think it should be made clearer that the most important insights in the paper come from the computer simulation of the mathematically intractable versions of the model. (Assuming the authors agree with this.)*

We realize that the decision of whether 'reality' lies in the simulations or in the analytical calculations is somewhat arbitrary, and that our choice of starting with the theory may seem a less straightforward approach to the problem we tackle. However, we do not agree that the main result of the paper lies in the simulations, because such simulations describe a specific, and very simplified, way groups can form, which is based on attachment. On the other hand, the analytical results allow to have a more general view of how group formation in general can affect the evolution of adhesiveness.

The results of the simulations do not appear to us particularly more crucial than those of the analytical model. It was maybe not clear that the individual-based model require, on average over several realizations, a minimum threshold of adhesiveness, exactly like the analytical solution for infinite population size. However, finite-size fluctuations can occasionally let the system overcome such a threshold, triggering evolution towards higher adhesiveness. This effect is well known whenever individual-based implementations of games are considered, and we think it does not require specific emphasis. On the other hand, what simulations provide is a confirmation that some assumptions of the adaptive dynamics framework do not have to hold strictly for achieving the results of the exact model. Moreover, numerical simulations allow to explore different choices for the fitness functional forms, that indeed bring new evolutionary outcomes.

We hope that the role of the simulations in supporting the conclusions of the paper is more explicit in the revised version of the manuscript. We have advanced to subheading “Group formation based on attachment” the description of the aggregation model base on differential attachment. We have dedicated subheading “Individual-based simulations” to the numerical exploration of the adaptive dynamics when group formation follows this scheme, and discussed the robustness with regard to theoretical results.

*3) It is great that the authors talk about adhesiveness as a continuous trait, especially because this turns the very binary problem of cooperation-defection or social-asocial into a much more nuanced question. But despite posing the problem so generally, the authors still use the social-asocial language, even when it doesn't make much sense. For example in paragraph five, subheading “Social behavior along evolutionary trajectories: altruistic sociality”, where both the resident and the mutant are actually sticky but one is a tiny bit less so and is then called asocial. That seems confusing (despite the fact that many people do this) and we would suggest finding a clearer way to refer to these traits that fits with the continuum perspective which, again, I think is a very good one.*

We agree that we should have been more careful about semantics, and have corrected the manuscript according to the referees' suggestion.

*4) The difference between cooperation and altruism has always been confusing. Indeed the way the authors present it is one way in which it has been described but not the only one and this part of the manuscript and its conclusions can lead to confusion. It seems that a lot of what people have termed cooperation and studied in the context of the Prisoner's dilemma is altruism according to the author's definition (see e.g. Nowak et al. work). And in many works altruism has been defined as an extreme form of cooperation, such as suicide (stalk cells in D. discoideum or worker ants in eusocial colonies). The authors don't seem to be talking about the latter when they're saying that altruism can evolve first. They seem to be talking about the former, which most people would interpret as actual cooperation (as opposed to mutually beneficial behavior). We are perfectly aware that the semantics of this field is a mess, but one ought to be very careful in phrasing the conclusions of this part of the study. Perhaps moving away from cooperation-altruism and phrasing it more generally and in more nuanced ways would help. Another possibility, if the authors want to stick with cooperation-altruism, is to explain the different definitions and cite other literatures as well. Because the results of this section seem very similar to, for example, people claiming that cooperation (rather than altruism) can evolve on networks if b/c > k.*

We tried our best to spell out in subheading “Social behavior along evolutionary trajectories” the point we wish to make about 'altruism', that is an extension to pre-existing definitions to the case when group size is distributed and not fixed. The idea is to keep the same concept as in the referenced literature, that altruism defines behaviors for which costs are not immediately recovered by marginal benefits from the cooperator's own contribution. Our extension merely relies on averaging such marginal benefits according to the resident population's group size distribution. With this in mind, we re-wrote the whole paragraph. We also referred to alternative definitions considered in the literature, that would actually be even less restrictive for altruism (namely, Kerr et al., TREE 2004).

The case of altruistic suicide is in our view just a limit case that is encompassed by other definitions of altruism. Moreover, in microbes it is not clear whether the 'strategy' is to commit suicide with certainty, or with a probability less than one. The latter case would result in a high, but finite effective fitness cost. We have specified in subheading “Social behavior along evolutionary trajectories” that altruistic suicide is a limit case of extremely high personal cost, that can however, in principle, be counterbalanced by very large benefits provided by the public good (for instance, when group cohesion is a fundamental determinant of survival).

There is no doubt that all conditions b/c>something are somehow alike. This is however the consequence of assuming that costs and benefits have additive effects on fitness. It is also not surprising that this 'something' is a statistics related to some aspect of the population structure (relatedness, connectedness of a network, number of individuals in groups, etc.; see for instance Nowak 2006, Science). Making explicit reference to all other cases where a condition similar to this hold, would result in increasing further a manuscript that is already quite long. We therefore opted for a more thorough explanation of how altruism can be defined based on the evaluation of the average marginal costs and benefits of an individual in a population of given structure.

*5) There is a related philosophical quibble concerning the distinction between net direct benefits (sometimes called weak altruism) and net costs (sometimes called strong altruism). If a social behavior gives a benefit to the actor that is greater than its cost, this may seem to be substantially different than the case where the actor's cost is greater than any benefit it receives. But it is relative fitness that matters and if the actor's social behavior benefits all group members, but others do not pay the cost borne by the actor, then there is no substantial difference. For simplicity assume that the actor provides benefit B to the group at cost C to the actor. If this benefit is split evenly among group members (including the actor) then actors (weak altruists) get B/n - C whereas other group members get B/n. Whether B/n - C is positive or negative doesn't affect the fact that actors in a group are at a disadvantage compared to non-actors in the group. Whether actors (altruists) are favored depends on population structure (assortment), not whether B/n - C is positive or negative. Weak altruists can be selected against if the interaction structure is unfavorable and strong altruists can be selected for if it is favorable. Of course the degree of assortment needed can be affected by the values of B and C (and therefore by whether B/n -C is positive or negative), but this is a matter of degrees of assortment, not a fundamentally different mechanism. Comparing two scenarios where B/n - C values are 2 and 6 respectively is not inherently different from comparing two scenarios where the values are -2 and +2 respectively. So it is not surprising or "counter-intuitive" that strong altruism can proceed weak altruism in the evolutionary trajectories of the model. The amount of assortment simply favors each in its turn.*

We agree with this, and this is actually the point we intended to make. In the context of our model, what happens in addition to the referee's observation is that the degree of assortment is coupled to the value of the trait (independently of b and c).

Even if this is somehow trivial if one thinks that average costs and benefits are affected by the population structure, we think that this is not often acknowledged in the literature on social behavior in microbes. Indeed, 'private goods' – providing direct benefits – are opposed to 'public goods', as if the assessment of costs and benefits could be disentangled from the interaction topology. On the theoretical side, weak altruism has been considered more easily achievable with respect to strong altruism.

From this, one may (but does not need to) conclude that the evolution of highly sacrificial forms of cooperation follow the establishment of directly beneficial forms of cooperation.

We think that the example we discuss shows that not only the social status depends on population structure, but also that the ease in attaining apparently different forms of cooperation cannot be determined if the social context changes on the same time scale as the trait underpinning the behavior.

We hope that in the revised version of the manuscript these messages are more clearly conveyed.

*6) As for the conclusion that the status of a social mutation can change along an evolutionary trajectory, that's a great point and an interesting conclusion. However, the conditions under which it happens seem very restrictive (the b/c ratio has to be very high). The authors should discuss the intuition and biological feasibility of such a scenario.*

According to our analysis, changes in the status of social mutations along the evolutionary trajectory appear in the parameter region b/c>T. While this might seem at first sight a restrictive condition on b/c, one has to keep in mind that it is also the delimiting inequality of directly beneficial vs. altruistic cooperation in standard PGG models occurring in groups of fixed size T. Here, altruism may occur for even larger values of b/c because, contrary to fixed group-size models, a proportion of individuals remain alone as long as the adhesiveness level is smaller than 1. Therefore, marginal returns from sociality (that occur only in groups) are not obtained systematically while the cost is always paid, whether or not in a group. The delimiting b/c value for altruism (1/r_alt_ in the manuscript) even diverges when adhesiveness is close to 0 as nobody gets in a group in such a case. As a consequence, a social mutation is always altruist for very small resident values of the adhesiveness.

In addition, it is not straightforward to relate game-theoretic parameters of cost and benefits such as in the PGG to actual biological costs and benefits (affecting for instance exponential growth rates in isolated populations). It is thus hard to tell what is realistic. Maybe having a large return from the collective phase, or a low cost for adhesion, is realistic. We preferred to avoid advancing hypotheses on the degree of realism of different levels of benefit-to-cost ratios, and take as a reference the theoretical condition that is commonly used for populations structured in groups of equal size.

We have however added a sentence to explain our first point at the end of subheading “Social behavior along evolutionary trajectories” (paragraph nine).

[Editors' note: further revisions were requested prior to acceptance, as described below.]

*In reading the response to reviewers on item 6 I was struck by the detail that in calculating the benefits and costs across the population, that all individuals (including loners) paid the cost of adhesiveness, while only those that made it into groups experienced the benefits. (I missed that detail in the first version, and its role in determining b/c ratios.) I don't think I understand the justification for this. If an extracellular matrix was the trait that caused adhesiveness and also benefited other group members, then loners might get at least some benefit from their own excretions of extracellular matrix. Also, one could imagine a different mechanism of adhesion that did not manifest itself unless a group was joined. In this case loners would not pay the cost. I would leave it up to the authors to decide whether providing some intuition behind this analysis decision is helpful.*

We added a few sentences to subheading “Life cycle and population structuring” to reply to Reviewer #3's comments stating that, for individuals endowed with the adhesive trait:

Adhesion costs might only be considered when they join a group;

Direct benefits might be retrieved even when they are ungrouped.

"The cost of adhesiveness is here assumed to be context-independent, thus it does not change conditionally to individuals belonging or not to a group […] would relax the social dilemma and promote even more efficiently increased adhesion."